# Crystal Structures of Fusion Cores from CCoV-HuPn-2018 and SADS-CoV

**DOI:** 10.3390/v16020272

**Published:** 2024-02-09

**Authors:** Fulian Wang, Guang Yang, Lei Yan

**Affiliations:** 1Shanghai Institute for Advanced Immunochemical Studies, ShanghaiTech University, 393 Middle Huaxia Road, Shanghai 201210, China; wangfl@shanghaitech.edu.cn; 2School of Life Science and Technology, ShanghaiTech University, 393 Middle Huaxia Road, Shanghai 201210, China

**Keywords:** coronavirus, post-fusion core, X-ray structure, SADS-CoV, CCoV-HuPn-2018

## Abstract

Cross-species spillover to humans of coronaviruses (CoVs) from wildlife animal reservoirs poses marked and global threats to human and animal health. Recently, sporadic infection of canine coronavirus–human pneumonia-2018 (CCoV-HuPn-2018) in hospitalized patients with pneumonia genetically related to canine and feline coronavirus were identified. In addition, swine acute diarrhea syndrome coronavirus (SADS-CoV) had the capability of broad tropism to cultured cells including from humans. Together, the transmission of Alphacoronaviruses that originated in wildlife to humans via intermediate hosts was responsible for the high-impact emerging zoonosis. Entry of CoV is mainly mediated by Spike and formation of a typical six helix bundle (6-HB) structure in the postfusion state of Spike is pivotal. Here, we present the complete fusion core structures of CCoV-HuPn-2018 and SADS-CoV from Alphacoronavirus at 2.10 and 2.59 Å, respectively. The overall structure of the CCoV-HuPn-2018 fusion core is similar to Alphacoronavirus like HCoV-229E, while SADS-CoV is analogous to Betacoronavirus like SARS-CoV-2. Collectively, we provide a structural basis for the development of pan-CoV small molecules and polypeptides based on the HR1-HR2 complex, concerning CCoV-HuPn-2018 and SADS-CoV.

## 1. Introduction

The coronaviruses, enveloped, positive-sense and single-stranded RNA viruses, are a large group of viral pathogens with a wide host range [1,2,3]. They can infect humans, other mammals, and birds and cause respiratory, hepatic, enteric and neurological diseases with varying severity [4,5]. Human CoVs (HCoVs) are often of animal origin and most of them originated from bats and then adapted to humans by direct jumping or by jumping into an intermediate species [6,7]. Bats are considered to be the primary reservoir of Alphacoronavirus and Betacoronavirus genuses and play a key role in interspecies transmission of coronaviruses [8]. To date, seven different strains of CoVs have been found to infect humans, of which three of them, SARS-CoV-1, SARS-CoV-2, and MERS-CoV, have caused pandemics, and pose severe threats to global health [9,10,11]. The SARS coronavirus is the best example, which has been proven to originate from the Chinese horseshoe bat and may be directly transmitted to humans or through intermediate hosts, such as civets [12,13]. In fact, the MERS coronavirus may also undergo spillover from bats to human through camels [14,15]. Meanwhile, coronaviruses infecting domestic animals also bring substantial economic losses and the risk of transmission to humans [16,17].

CCoV-HuPn-2018 is the human isolate of canine CoV (CCoV), which was isolated from Malaysian patients with pneumonia, experiencing a respiratory illness [18]. To date, this is the eighth known coronavirus that can infect humans and cause disease after SARS-CoV-2, which has attracted extensive attention worldwide [19]. Another swine acute diarrhea syndrome coronavirus (SADS-CoV) is a newly discovered Alphacoronavirus that causes watery diarrhea and acute vomiting mainly in piglets, in southern China [20,21]. It is demonstrated that SADS-CoV has a broad host range and has inherent potential to disseminate between animal and human hosts, perhaps using swine as an intermediate species and becoming a threat to human health [22,23]. Therefore, for SADS-CoV, which has the potential to infect humans and CCoV-HuPn-2018, it is important to investigate the structure and function to aid in the development of preventative or therapeutic strategies.

Virus entry of CoV is mediated by its spike (S) protein, where its S1 domain facilitates attachment to host cells, and its S2 domain is involved in the subsequent fusion between the virus and host membrane [24,25]. During the fusion process, two heptad repeats, HR1 and HR2, in the S2 domain, assemble into a six-helix membrane fusion structure termed the fusion core [26,27]. In this study, we presented a 2.10 Å and 2.59 Å crystal structure of the viral fusion cores of CCoV-HuPn-2018 and SADS-CoV. Three HR1 helices form the central coiled coil by hydrophobic interactions of the HR repeats. HR2, thereafter, binds to the side groove formed by HR1 chains. We noted an extensive contact network between HR1 and HR2, involving both hydrophobic and H-bond interactions. In addition, the side groove of the HR1 coiled coil is deep for its N-terminal part and relatively shallow for the C-terminal part, complementarily accommodating the helical and extended-loop halves of HR2, respectively. Therefore, our work sheds light on the molecular mechanism of membrane fusion of CCoV-HuPn-2018 and SADS-CoV and provides the structural basis for the binding potential of pan-CoV entry inhibitor EK1 to HR1s from CCoV-HuPn-2018 and SADS-CoV.

## 2. Materials and Methods

### 2.1. CoV Sequence Alignment and Phylogenetic Analysis

Sequence alignment was completed using online tools of Clustal Omega and ESPript. Phylogenetic tree was constructed using the MEGAX using sequences mainly downloaded from the UniProt website. Accession numbers used for phylogenetic analysis are as follows: SARS-CoV-2 (P0DTC2), SARS-CoV-1 (P59594), HCoV-229E (P15423), HCoV-NL63 (Q6Q1S2), HCoV-HKU1 (U3NAI2), HCoV-OC43 (U3NAI2), MERS-CoV (R9UQ53), TGEV (P07946), Bat SARS-CoV Rf1/2004 (Q0QDZ0), Feline CoV (C5IGD1), BtCoV-BM48-31/BGR/200 (E0XIZ3), SADS-CoV (A0A2P1G7F5), BtCoV-HKU2 (A8JNZ2), Feline Infectious Peritonitis Virus or FIPV (Q66951), Bovine CoV or BCoV (P25191), Porcine deltacoronavirus (W8Q9Y7), Munia coronavirus HKU13-3514 (B6VDY7), MHV-A59 (P11224), BtCoV-HKU4 (A3EX94), BtCoV-HKU5 (A3EXD0), SARS like CoV WIV1 (U5WI05), SARS like CoV Rs3367 (U5WHZ7), BtCoV-HKU9 (A3EXG6), Avian infectious bronchitis virus or IBV (P11223), Beluga whale coronavirus SW1 (B2BW33), Bottlenose dolphin HKU22 (V5TFD8), RaTG13 (A0A6B9WHD3), BtCoV-HKU3 (Q3LZX1), Civet SARS-CoV 007/2004 (Q3ZTF3), Canine coronavirus or CCoV (D2WXL7) and Sparrow deltacoronavirus or SpDCoV (A0A2Z4EVU6_9NIDO). Pangolin coronavirus GX/P2V/2017 (QVT76606.1) and GD/1/2019 (QLR06867.1), Hu-PDCoV (MW685622), CCoV-HuPn-2018 (MW591993), RpYN06 (MZ081381), RmYN02 (MW201982), BANAL-20-247/Laos/2020 (MZ937004), BANAL-20-52/Laos/2020 (MZ937000) and Montifringilla taczanowskii coronavirus or MtCoV (MT215336) were downloaded from NCBI website. RshSTT182 (EPI_ISL_852604) is available from the GISAID database.

### 2.2. Plasmid Construction

To express the fusion cores of SADS-CoV, the coding sequences for HR1 (residues 764–842) and HR2 (residues 1025–1061) were connected via a short linker (encoding L6: SGGRGG) by overlapping PCR. The resulting sequence of HR1-L6-HR2 was then subcloned into the pET-28a vector with an N-terminal in-frame SUMO-tag. The similar method was also applied to construct plasmid of HR1-L6-HR2 (CCoV-HuPn-2018).

### 2.3. Protein Expression and Purification

For protein expression, the pET28A-SUMO-HR1-L6-HR2 (CCoV-HuPn-2018) plasmid was transformed into Escherichia coli strain BL21 (DE3) competent cells. A single colony was inoculated into 4 mL Luria–Bertani (LB) medium containing 100 μg/mL of kanamycin (Sangon, Shanghai, China) and incubated overnight at 37 °C. The overnight culture was then seeded into two liters of fresh LB medium and cultured at 37 °C until the OD_600_ (optical density at 600 nm) reached 0.6. Target protein overexpression was then induced with 1 mM isopropyl-D-thiogalactoside (IPTG, AMRESCO, Solon, OH, USA) at 16 °C overnight. After harvesting via centrifugation at 20,000 rpm for 30 min, the cell debris was re-suspended in lysis buffer (50 mM TRIS-HCl pH 7.5, and 300 mM NaCl) supplemented with phenyl methyl sulfonyl fluoride (PMSF, Biovision, Milpitas, CA, USA), and lysed using a sonicator (Thermo Fisher Scientific, Waltham, MA, USA). The cell lysate was then centrifuged at 70,000× *g* for 20 min at 4 °C. The supernatant was collected and loaded onto TALON metal affinity resin (Clonetech, Mountain View, CA, USA). After extensive washing, the HR1-L6-HR2 protein of interest was eluted with lysis buffer supplemented with 500 mM imidazole. The fractions eluted from the TALON column were then dialyzed against dialysis buffer (20 mM TRIS-HCl pH 8.0, 5 mM β-ME and 150 mM NaCl) overnight at 4 °C and then processed with Ulp1 enzyme (at 1:100 *w*/*w* ratio) to remove the SUMO tag from HR1-L6-HR2. Finally, the cleaved product was reloaded to TALON resin to remove Ulp1 and His-SUMO. The flow-through was then applied to a Superdex-75 gel filtration column (Cytiva, Little Chalfont, Buckinghamshire, UK). Fractions containing homogeneous HR1-L6-HR2 trimer were collected and concentrated by ultrafiltration (Satorious, Göttingen, Germany). The similar method was also applied to construct plasmid of HR1-L6-HR2 (SADS-CoV).

### 2.4. Protein Crystallization

The HR1-L6-HR2 fusion protein was crystallized at 20 °C using the hanging drop, vapor-diffusion method. Crystals were grown on a siliconized cover clip by equilibrating a mixture containing 1 μL protein solution (10 mg/mL HR1-L6-HR2 trimer in 20 mM Tris-HCl pH 8.0, 150 mM NaCl) and 1μL reservoir solution (0.1 M HEPES pH 7.5, 42% PEG200) for SADS-CoV and (0.1 M TRIS pH 8.5, 25% PEG4000) for CCoV-HuPn-2018 against 500 μL reservoir solution. After about 5 days, single crystals grew out and were then flash-cooled in liquid nitrogen after adding 20% glycerol as cryo-protectant for data collection.

### 2.5. Data Collection and Structure Determination

Diffraction data were collected at beamline BL02U1 (previously known as BL17U1) and BL19U1 of the Shanghai Synchrotron Radiation Facility (SSRF) [28]. Raw data were indexed and processed using HKL3000 [29] and molecular replacement was performed with the Phaser program in PHENIX [30]. The initial model was further improved by cycles of manual building and refinement using COOT [31] and PHENIX [30]. Data collection and refinement statistics are shown in Table 1. Atomic coordinates and structure factors have been deposited in the Protein Data Bank (PDB entry: 8X7X and 8X7Z). Figures were prepared using program Pymol (http://www.pymol.org, accessed on 15 November 2023). Electrostatic calculations were performed in Pymol with PDB2PQR plug-in [32].

### 2.6. Homology Model Building

Models for HR1(CCoV)-EK1 and HR1(SADS)-EK1 were derived through homology modeling using Swiss Model website [33]. The template for HR1(CCoV)-EK1 and HR1(SADS)-EK1 were obtained from the crystal structures of HR1(SARS)-EK1 and HR1(229E)-EK1 [34]. To relax and stabilize the interaction between EK1 and corresponding HR1, initial models were optimized by performing energy minimization, followed by a 5-ns molecular dynamics simulation using Schrödinger Suite (www.schrodinger.com, accessed on 20 November 2023). The simulation systems were solvated with full-atom TIP3P water, containing Cl^−^ and Na^+^ ions at a concentration of 0.15 M to mimic physiological ionic strength. During the simulation, temperature T and pressure P were kept constant at 310 K and 1 atm, respectively.

## 3. Results

### 3.1. Phylogenetic Analysis of Multiple CoVs According to Spike Sequences Including CCoV-HuPn-2018 and SADS-CoV

Like HCoV-229E and HCoV-NL63, SADS-CoV and CCoV-HuPn-2018 are also members of Alphacoronaviruses in the family Coronaviridae (Figure 1a). Sequence alignment of HCoVs indicates that a 14-amino-acid insertion arises in the HR1 regions of α-CoV including HCoV-NL63 and HCoV-229E and CCoV-HuPn-2018 that would make the HR1 helix of Alphacoronaviruses four turns (two heptad repeats) longer than those of β-CoV MERS-CoV or SARS-CoV-2 (Figure 1b). However, SADS-CoV lacks a 14-residue insertion in HR1, which is different from other porcine-associated CoVs that include α-CoV porcine transmissible gastroenteritis virus (TGEV), porcine respiratory coronavirus (PRCV) and porcine epidemic diarrhea virus (PEDV) or δ-CoV porcine deltacoronavirus (PDCoV) (Figure 1c).

### 3.2. Generation of Fusion Cores of CCoV-HuPn-2018 and SADS-CoV

Here, we designed a linked recombinant SADS-CoV S2 fusion protein (HR1-HR2) using N-terminal residues 764–842 and C-terminal residues 1025–1061, with a six-amino-acid linker (L6: SGGRGG) between the two regions (Figure 2e). Likewise, HR1 (residues 1059–1141) and HR2 (residues 1326–1376) regions of CCoV-HuPn-2018 S were constructed (Figure 2a). Similar linkers have been utilized in structural studies of MERS-CoV and HCoV-229E fusion cores and found not to affect the intrinsic interaction (or packing) between HR1 and HR2 [35,36].

### 3.3. Overall Architecture of Fusion Cores of CCoV-HuPn-2018 and SADS-CoV

HR1-L6-HR2 (CCoV-HuPn-2018) recombinant protein crystallized in space group R32:H, with cell parameters a = b = 45.6 Å, c = 428.7 Å and angles of 90°, 90° and 120° (Table 1). The structure was solved by molecular replacement, using the crystal structure of the HCoV-229E fusion core (PDB entry: 5YL9) as the search model and refined to a final resolution of 2.10 Å with R_work_ of 24.0% and R_free_ of 29.0% (Table 1). HR1 of CCoV-HuPn-2018 forms a 23-turn α-helix, while HR2 adopts mixed conformations: residues T1342-T1372 fold into a nine-turn α-helix while residues V1326-L1341 and L1373-L1376 on either side of the helix adopt an extended conformation (Figure 2b). Then, three HR2 helices pack in an oblique, left-handed and antiparallel direction into hydrophobic grooves on the surface of this trimeric coiled coil, forming in a six-helical bundle structure with dimensions of ~116 Å in length and 30 Å in diameter with a left-handed supercoil (Figure 2b). HR1 from three HR1-L6-HR2 (CCoV-HuPn-2018) molecules are arranged around the crystallographic three-fold symmetry axis and form the central hydrophobic core (Figure 2c). To compensate for their longer HR1 regions, the HR2 region of CCoV-HuPn-2018 also contains a 14-residue insertion that folds to a longer HR2 helix (Figure 2b). Overall, the fusion core structure of CcoV-HuPn-2018 is very similar to other Alphacoronaviruses, such as HcoV-229E and HcoV-NL63 (Appendix A).

The HR1-L6-HR2 (SADS-CoV) construct also crystallized in the space group R32:H, using the crystal structure of the SARS-CoV-2 fusion core (PDB entry: 6LXT) as the search model, with cell parameters a = b = 45.2 Å, c = 418.0 Å and angles of 90°, 90° and 120° (Table 1). One copy of the HR1-L6-HR2 molecule was shown in an asymmetric unit. The structure was refined to a final resolution of 2.59 Å with R_work_ of 23.4% and R_free_ of 28.2% (Table 1). HR1 of SADS-CoV forms a 19-turn α-helix trimer, while these residues first form the random coil tether (residues 1025–1036) and then a second heptad repeat region (HR2) between residues 1037–1050, followed by the other extended conformation residues 1051–1061 (Figure 2f). HR1s from three HR1-L6-HR2 (SADS-CoV) molecules are arranged around the crystallographic three-fold symmetry axis and also form the central hydrophobic core in an antiparallel manner, resulting in a six-helical bundle structure with dimensions of ~107 Å in length and 30 Å in diameter with a left-handed supercoil (Figure 2g). The HR2 helices make five full turns and pack into the central HR1 trimer to form a highly stable six-helix bundle conformation that coordinates the fusogenic events between the virus and host cell membranes (Figure 2h). Such a structure is typical for the CoVs fusion core and represents the post-fusion state of the SADS-CoV spike protein. Overall, the fusion core structure of SADS-CoV is similar to those reported for Betacoronaviruses with only 5 helix turns in HR2 (Appendix A).

A structural comparison between SADS-CoV and SARS-CoV-2 fusion cores revealed a high degree of overall structural similarity with a root mean squared deviation (RMSD) of approximate 0.814 Å for 71 Cα atoms of a piece of HR1 protomer (Appendix A left panel). And the superimposition of the two structures of CCoV-HuPn-2018 and HCoV-229E revealed an RMSD of ~0.477 Å for 83 Cα atoms of HR1 (Appendix A right panel). All of the helical elements and a majority of the extended HR2 loops could be well aligned. Only some terminal residues exhibit conformational variance (Appendix A).

### 3.4. Hydrophobic Packing between HR1-HR2 of CCoV-HuPn-2018 or SADS-CoV

Hydrophobic residues are vital for stability of the S2 postfusion state [37]. A total of observed 52 or 35 HR2 residues of CCoV-HuPn-2018 and SADS-CoV S, in both extended and helical conformations, spanning about 110 Å and 80 Å, matching most of the length of the HR1 trimer (Figure 3a,b). Specifically, taking the CCoV-HuPn-2018 fusion core for example, HR2 residues 1342–1372 of CCoV-HuPn-2018 fold into a nine-turn amphipathic α-helix, which fits snugly onto the 3HR1-core mainly through extensive hydrophobic interactions (Figure 3d). It was observed that concerning the hydrophobic groove formed by HR1 helices, the N-terminal half is relatively deep, accommodating the C-terminal helix of HR2. In contrast, the C-terminal half of the groove is shallow and accommodates the HR2 extended N-terminal loop. Amongst, side chains of hydrophobic HR2 residues (Leu-1329, Leu-1331, Phe-1334, Leu-1339, Leu-1341 and Leu-1348) are completely buried in the cavities on the 3HR1 grooves (Figure 3d). Besides the nine-turn helix, between the N-terminal Val-1326 and Leu-1341, plus from the C-terminal Thr-1372 to Leu-1376, HR2 has an extended conformation, contacting HR1 residues 1101–1134 and 1059–1068, respectively (Figure 3d). Among them, filled by the side chains of Leu-1339 and Leu-1341 (Appendix A), could be a potential pocket for small molecule inhibitor targeting [27]. Therefore, in both CCoV-HuPn-2018 and SADS-CoV fusion core structures, the HR1 and HR2 polypeptides are highly complementary in both shape and chemical properties (Figure 3c,e).

### 3.5. Electrostatic and Polar Stapling between HR1 and HR2 of CCoV-HuPn-2018 or SADS-CoV

Besides hydrophobic packing, electrostatic and polar interactions also contribute immensely to the affinity and specificity between HR1 and HR2 (Figure 4). Only three-pair polar side-chain to side-chain interactions were detected between the helical region of HR2 and HR1 of SADS-CoV (Figure 4b). On the contrary, in the helical region of CCoV-HuPn-2018, residues on HR2 mainly engage residues on HR1 through side-chain to side-chain interactions via not only hydrogen bonds but also salt bridges (Figure 4a). In the extended region of HR2, extensive polar interactions were observed between the main-chain atoms on HR2 and the side-chains of residues from HR1 (Figure 4c,d). It is worth noting that some of these interactions are highly conserved among CCoV-HuPn-2018 and SADS-CoV even across different HCoVs (Figure 4c,d, conserved interactions are underlined). All of these hydrogen-bond interactions between side chains of polar HR1 amino acids and main-chain nitrogen and oxygen atoms of HR2 help constrain the conformation of this extended part of HR2.

### 3.6. Implication for Targeting HR1 Region of CCoV-HuPn-2018 or SADS-CoV by Pan-CoV Fusion Inhibitor

Side-by-side electrostatic surface comparisons also reveal similarities and differences between α-CoVs like SADS-CoV, CCoV-HuPn-2018 and HCoV-229E. The electrostatic surface of binding sites of helix region of HR2 of SADS-CoV and CCoV-HuPn-2018 are quite similar to each other (boxed in a blue dashed line) and, nonetheless, opposite to HCoV-229E (Appendix A). The upper and lower areas share more similarities including hydrophobicity among these three α-CoVs, which usually accommodate bulky hydrophobic residues and registration of extended regions of HR2 (Appendix A). In the past, collaborating with our cooperators, we have developed a pan-CoV fusion inhibitor EK1 successfully [34]. Accordingly, extensive and highly conserved hydrophobic and hydrophilic interactions between EK1 and 3HR1 cores endow EK1 with the ability to bind the 3HR1 cores from different HCoVs and, hence, the capability of blocking the association of HR2s onto their cognate 3HR1 cores. With a view to SADS-CoV and CCoV-HuPn-2018, in this pioneer comparative structural study, we carried out homology modeling to probe the binding potential of EK1 to HR1s of these two Alphacoronaviruses. Models of HR1(CCoV)-L6-EK1 and HR1(SADS)-L6-EK1 illustrated that EK1 would be able to form similar hydrophobic (Figure 5a,b) and hydrophilic interactions (Figure 5c,d) with HR1s, as we observed in the structure of HR1(229E)-L6-EK1. This evidence further indicated that the structural plasticity and electrostatic compatibility acclimatize EK1 to the different geography of the hydrophobic groove of the 3HR1 core. However, the validation of authentic efficacy of EK1 against infection of CCoV-HuPn-2018 and SADS-CoV also need further experiments.

## 4. Discussion

Entry into susceptible host cells is the first step in the virus life cycle, and each entry process starts with receptor recognition [38]. Adaptation to a new receptor on cells of the new host species is an important factor for crossing the species barrier. During evolution for a host jump, coronaviruses have developed a large diversity of binding specificities with high flexibility in the coronaviral spike protein [39]. Isolation of CCoV-HuPn-2018 from hospitalized patients with pneumonia and the introduction of an oligosaccharide at position N739 of the human aminopeptidase N (APN), rendering cells susceptible to CCoV-HuPn-2018 spike-mediated entry, suggest that this virus is a potential HCoV [19,40]. Even though SADS-CoV is an HKU2-related bat coronavirus, causing outbreaks of severe watery diarrhea of suckling piglets with a mortality of up to 90% in several commercial pig farms in Guangdong Province, China [20]. However, SADS-CoV was found to possess a wide range of host tropism and infect multiple cell lines originating from vertebrates including humans in the laboratory [23]. SADS-CoV also replicated efficiently in several different primary human lung cell types, as well as primary human intestinal cells, implicating SADS-CoV as a potential higher-risk emerging coronavirus pathogen that could negatively impact the global economy and human health [22]. All of them arouse concerns about Alphacoronaviruses.

So far, few effective therapies can be applied to prevent or cure the CCoV-HuPn-2018-related diseases. Receptor recognition and membrane fusion are two crucial processes initiating infections by enveloped viruses. The molecular basis of receptor binding by CCoV-HuPn-2018 has recently been illustrated [19]. But little is known about attachment and receptor integration of spike of SADS-CoV [41]. Here, we considered it to be of great interest to explore the fusion mechanism of these novel coronaviruses. In this study, we have determined the fusion core structures of CCoV-HuPn-2018 and SADS-CoV at 2.10 Å and 2.59 Å, respectively, thus further completing the structural profiles of α-HCoVs fusion cores. The solved structures show a typical six-helix bundle fold, as expected. There are three main genera (alpha, beta and gamma) in the Coronaviridae family [42]. CCoV-HuPn-2018 and SADS-CoV all belong to the Alphacoronavirus genus, as do HCoV-229E and HCoV-NL63 (Figure 1). Nevertheless, SADS-CoV is categorized into a phylogenetically independent subgroup lineage 1b*, in contrast with CCoV-HuPn-2018 which is classified into subgroups 1b. The CCoV-HuPn-2018 fusion core structures reveal that its HR1 folds into a 23-turn helix and HR2 comprises 9 turns. Intriguingly, owing to a lack of 14 aa insertions, SADS-CoV harbors 19 turns in HR1 (Figure 2). Consequently, HR2 motif of SADS-CoV has only five turns in the helix region, the first reported exception in α-HCoV genus (Figure 2). Therefore, this study also provided the first glimpse of the fusion core structures of subgroup 1b* coronaviruses.

Similar to other HCoVs, the high affinity interaction between HR1 and HR2 of CCoV-HuPn-2018 and SADS-CoV are mediated by both extensive hydrophobic and polar interactions (Figure 3 and Figure 4). Despite, side-by-side electrostatic surface comparisons of the 3HR1 hydrophobic core also reveal substantial differences between CCoV-HuPn-2018 and SADS-CoV. Some polar interaction modes are similar across different HCoVs. Together, these conserved interactions and the differences in electrostatic surface potential of the 3HR1 hydrophobic core across different HCoVs should be taken into consideration when designing pan-CoVs inhibitors that mimic HR2 for targeting HR1.

Given the high genomic mutation rate of CoVs, climate change, destruction of wildlife habitat and the growing global population migration, future outbreaks of CoV-mediated pandemics like SARS or MERS, from the Alphacoronavirus genus are very likely to occur [43,44,45]. A recent study demonstrated that the conserved HR1 domain in the S2 subunit, mimicking the fusion intermediate conformation, can serve as a novel target for the development of variant-proof SARS-CoV-2 or pan-sarbecovirus vaccines in vivo [46]. To this end, structural and functional characterization of fusion cores of CCoV-HuPn-2018 and SADS-CoV take on greater significance, especially for structure-based design of future therapies or vaccines.

## 5. Conclusions

The formation of a six-helix bundle (6-HB) fusion core brings the virus closer to the cell membrane and promotes membrane fusion and infection. In this paper, we presented high-resolution structures of fusion cores of CCoV-HuPn-2018 and SADS-CoV, elucidating the structural basis of membrane fusion mediated by HR1 and HR2. This information can be valuable in understanding the infection mechanism of the virus and provides new ideas for the design and development of antiviral drugs. We hope that this work will stimulate further work and efforts to advance the prevention and treatment of potential infections of zoonotic Alphacoronaviruses.

## 6. Limitations of Study

Here, we showed the high-resolution structures of fusion cores of CCoV-HuPn-2018 and SADS-CoV from α-CoV genus, which are critical for entry, and a side-by-side structural comparison between CCoV-HuPn-2018 and SADS-CoV and other α and β coronaviruses showed high similarity. Indeed, our study provided only structural information and homology models. Further studies will be needed to elucidate whether the interactions between HR1 and HR2 are vital for fusion and effects of fusion inhibitor EK1 against these two α-CoVs.

## Figures and Tables

**Figure 1 viruses-16-00272-f001:**
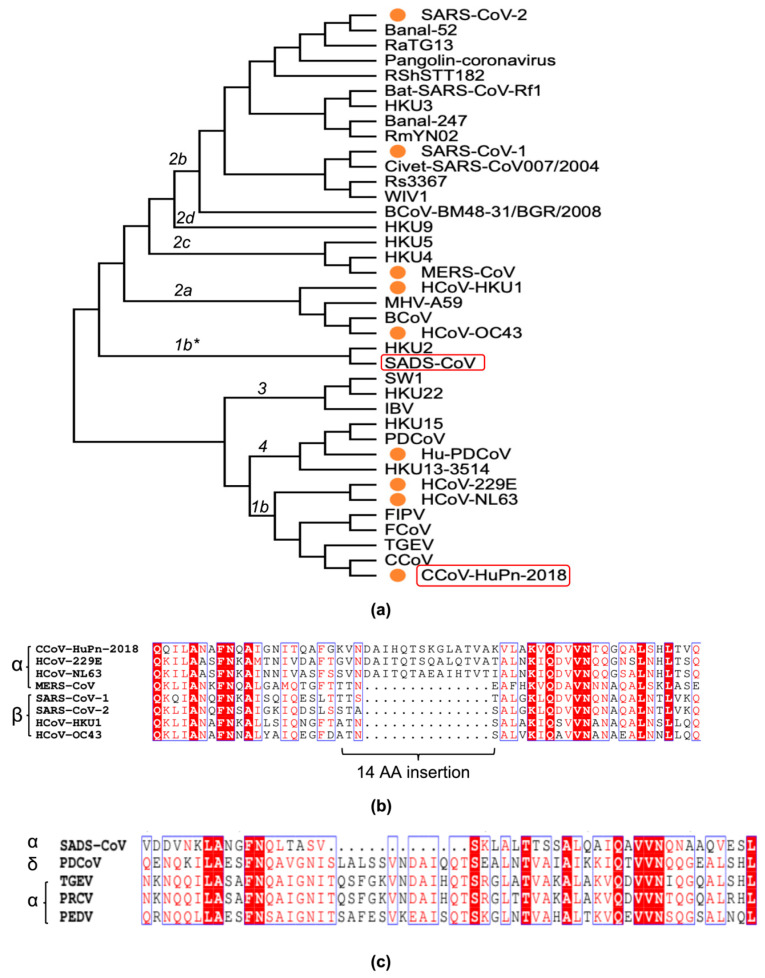
Phylogenetic analysis of some coronaviruses and sequence alignments of HR1 regions of SADS-CoV and CCoV-HuPn-2018 compared to related coronaviruses. (**a**) Phylogenetic tree of fusion proteins (S) from some related coronaviruses. SADS-CoV and CCoV-HuPn-2018 are boxed in red; (**b**) sequence alignment of HR1 regions of HCoV-HKU1, HCoV-OC43, SARS-CoV-1, SARS-CoV-2, MERS-CoV, HCoV-229E, HCoV-NL63 and CCoV-HuPn-2018 (14-aa insertion enclosed); (**c**) sequence alignment of HR1regions of SADS-CoV, PDCoV, TGEV, PRCV and PEDV.

**Figure 2 viruses-16-00272-f002:**
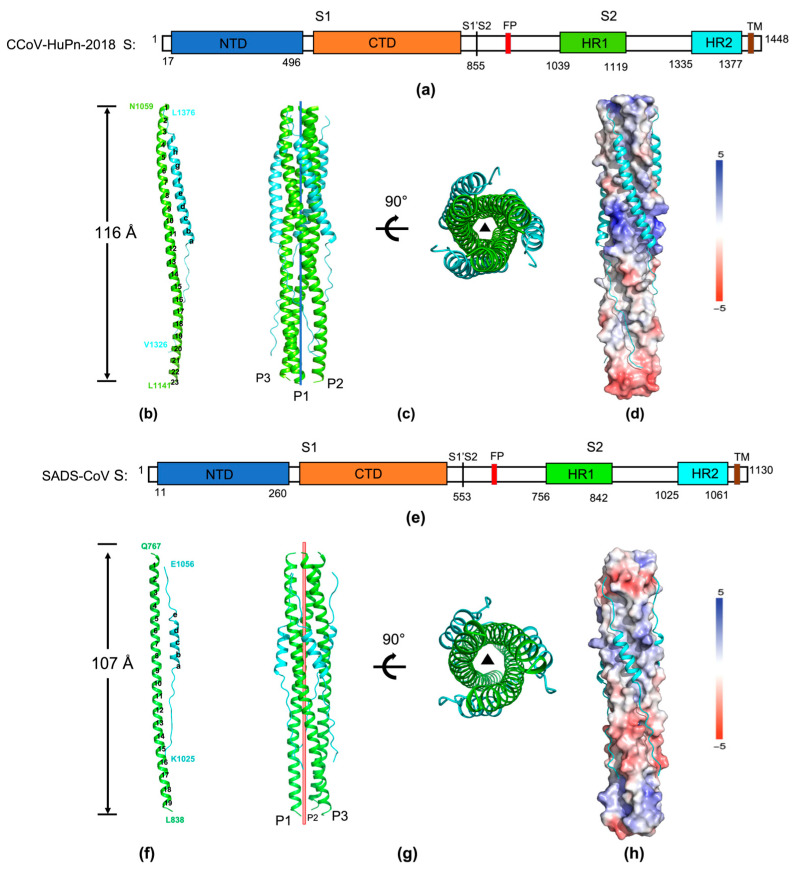
Overall structures of the CCoV-HuPn-2018 and SADS-CoV fusion core. (**a**,**e**) Cartoon characterization of CCoV-HuPn-2018 and SADS-CoV spike protein and prediction of the two heptad-repeat regions. NTD, N-terminal domain; CTD, C-terminal domain; S1 and S2, two cleaved fragments of the fusion protein; HR, heptad repeat region; FP, fusion peptide; and TM, transmembrane region; Ribbon representation of the S2 core structure in the form of monomer of CCoV-HuPn-2018 (**b**) and SADS-CoV (**f**) and trimer CCoV-HuPn-2018 (**c**) and SADS-CoV (**g**) fusion protein. Numbers of helix turns are labeled. (**c**,**g**) The red and blue pole (left panel) and the filled triangle (right panel) represent the trimer three-fold axis. (**d**,**h**) The hydrophobic groove between the 3-HR1 and HR2 of SADS-CoV (**h**) and CCoV-HuPn-2018 (**d**). Electrostatic potential surface of the HR1 inner coil trimer with HR2 peptides is shown as ribbon. The solvent-accessible surface is colored according to the electrostatic potential, which ranges from +5 V (most positive, dark blue) to −5 V (most negative, dark red) with hydrophobic in white.

**Figure 3 viruses-16-00272-f003:**
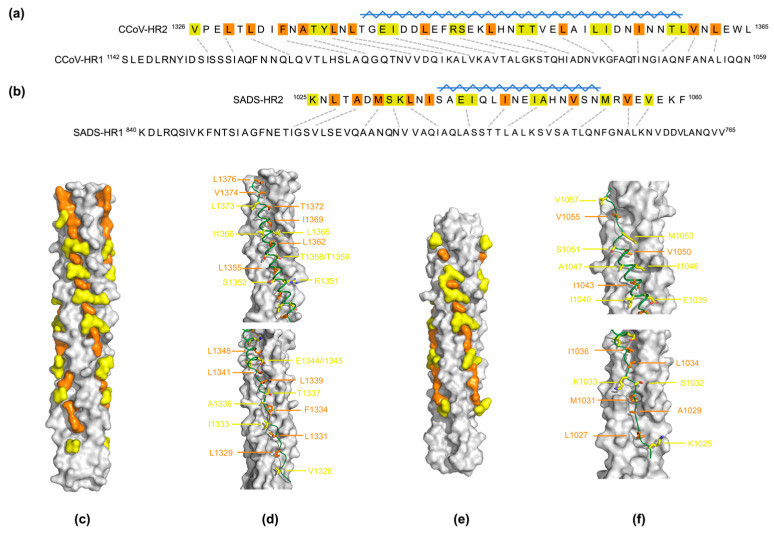
Two types of hydrophobic interactions are observed between the HR1 and HR2 helices packing geometry. HR1 residues of CCoV-HuPn-2018 (**a**) and SADS-CoV (**b**) fusion cores interacted with hydrophobic HR2 residues linked with dashed lines located in the same layers on the 3HR1 triple helix. Burying HR2 residues are shaded orange, and ridge-packing HR2 residues are shaded light yellow. Helix region of HR2 is indicated by light blue symbol. Surface representation of HR1 and HR2 helices illustrates that HR2 residues fit snugly onto the surface of 3HR1core of CCoV-HuPn-2018 (**c**) and SADS-CoV (**e**), thereby filling HR1 hydrophobic cavities and masking its hydrophobic surface. The 3HR1core is shown as an electrostatic surface, and HR2 residues involved in hydrophobic interactions are depicted in orange (completely buried) and yellow (packing, ~50% buried) surfaces, respectively; HR2 helices are shown as teal ribbons on the dark grey surface of the 3HR1 cores of CCoV-HuPn-2018 (**d**) and SADS-CoV (**f**). HR2 residues that bury their side chains completely into the cavities on HR1 are shown as orange stick models and HR2 residues that pack around 50% of the solvent accessible surface of their side chains on ridges of HR1 are depicted as yellow stick models.

**Figure 4 viruses-16-00272-f004:**
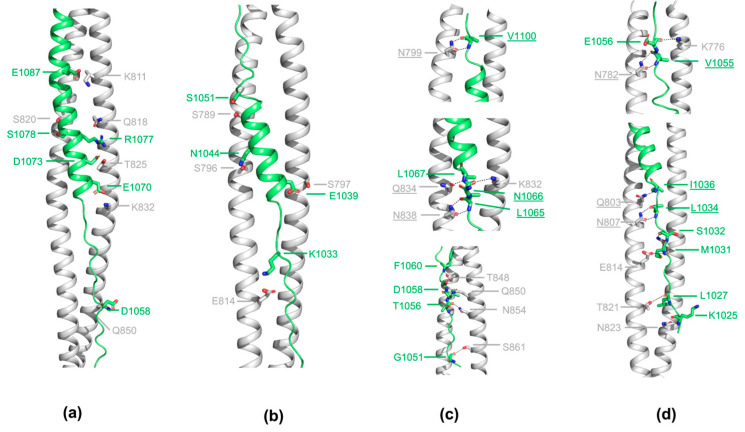
Interactions between side-chain residues of HR1 and side-chain or main-chain atoms of HR2. Side-chain to side-chain interactions between HR1 and HR2 of CCoV-HuPn-2018 (**a**) and SADS-CoV (**b**); interactions between side-chain residues of HR1 and main-chain atoms of HR2 are only seen in the N-terminal and C-terminal extended region of HR2 of CCoV-HuPn-2018 (**c**) and SADS-CoV (**d**). The HR1 motifs are depicted as white ribbons and the HR2 motifs as green ribbons. For clarity, only two HR1 motifs of the 3HR1 core are shown. The HR1 and the HR2 residues that are involved in interactions are illustrated as stick models, with black dash lines representing hydrogen bonds and/or salt bridges.

**Figure 5 viruses-16-00272-f005:**
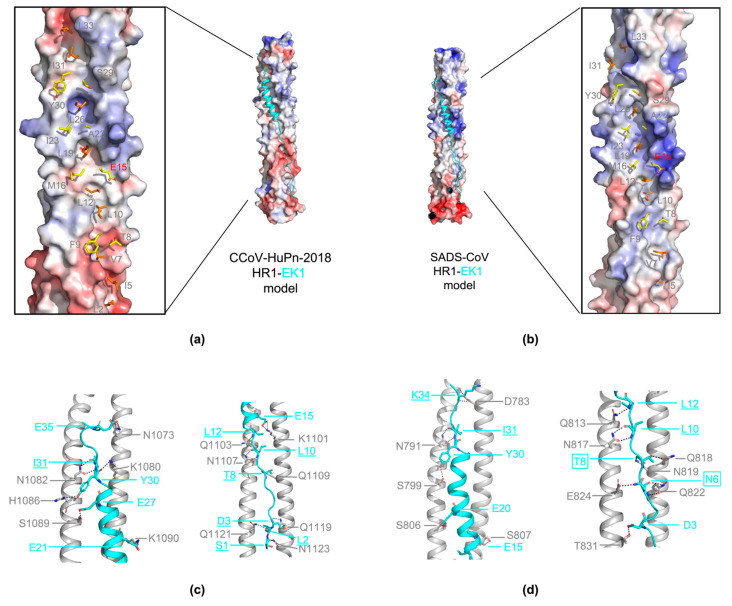
Model of HR1(CCoV)-EK1 (**a**) and HR1(SADS)-EK1 (**b**) also shown as electrostatic surface (3HR1) and ribbon (EK1, cyan). Close-up views display EK1 residues that bury over 70% of their side-chain solvent accessible surface (SAS) into these pockets shown as orange stick models at the positions where the hydrophobic surface on the 3HR1 core is deep concave (pockets). And at the locations where the hydrophobic surface on the 3HR1core is relatively flat (ridges), EK1 residues that pack 50 to 70% of their side-chain SAS against these ridges are shown as yellow stick models; putative hydrophilic interactions between EK1 with CCoV-HR1 (**c**) or SADS-HR1 (**d**). Residues participating in main-chain to side-chain interactions are underlined, and not only main-chain to side-chain but also side-chain to side-chain interactions are boxed.

**Table 1 viruses-16-00272-t001:** X-ray crystallographic data processing and refinement statistics.

	HR1-L6-HR2 (SADS-CoV)	HR1-L6-HR2 (CCoV-HuPn-2018)
**Data collection statistics**		
Beamline	SSRF-BL02U1	SSRF-BL19U1
Wavelength (Å)	0.9795	0.9789
Space group	R32:H	R32:H
Cell dimensions		
a, b, c (Å)	45.2, 45.2, 418.0	45.6, 45.6, 428.7
α, β, γ (°)	90, 90, 120	90, 90, 120
Resolution range (Å) *	28.56–2.59 (2.68–2.59)	71.45–2.10 (2.17–2.10)
No. of unique reflections	5590 (548)	10736 (1057)
Completeness (%) *	98.93 (99.09)	99.70 (100.00)
<I/σ(I)> *	7.6 (2.8)	5.8 (2.9)
CC_1/2_ *, a	0.984 (0.359)	0.974 (0.432)
Wilson B-factor (Å2)	51.26	10.91
**Refinement statistics**		
Reflections used in refinement *	5551 (547)	10712 (1057)
Reflections used for R_free_ *	285 (33)	502 (62)
R_work_ (%) *, b	23.4 (25.6)	24.0 (29.3)
R_free_ (%) *, b	28.2 (37.1)	29.0 (39.2)
No. of non-hydrogen atoms		
Protein	798	1047
Solvent	7	109
Average B-values (Å^2^)	75.58	19.16
Protein	75.68	17.75
Solvent	60.00	32.64
RMSD bond length (Å)	0.01	0.02
RMSD bond angle (°)	0.78	1.59
Ramachandran favored (%)	92.23	99.24
Ramachandran allowed (%)	7.77	0.76
Ramachandran outlier (%)	0	0
PDB ID	8X7X	8X7Z

* Statistics for the highest-resolution shell are shown in parentheses. a ${CC}_{\frac{1}{2}}=\frac{\sum (x-\langle x\rangle )(y-\langle y\rangle )}{{\left[\sum {\left(x-\langle x\rangle \right)}^{2}\sum {\left(y-\langle y\rangle \right)}^{2}\right]}^{\frac{1}{2}}}$. b R_work_ $=\frac{\sum \left|\left|F_{\mathrm{obs}}\right|-\left|F_{\mathrm{calc}}\right|\right|}{\sum \left|F_{\mathrm{obs}}\right|}$; R_free_ is defined as R_work_ calculated from 5% of the reflections that were excluded from refinement.

## Data Availability

Supplementary Material is available online. Raw data are available from the corresponding authors upon reasonable request. MTZ and PBD documents were deposited on the PBD website with entries of 8X7X and 8X7Z.

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
