# Peer review of "Crystal Structures of Fusion Cores from CCoV-HuPn-2018 and SADS-CoV"

_viruses, 2024, doi:10.3390/v16020272_

Round 1

Reviewer 1 Report

Comments and Suggestions for Authors

The submitted paper represents very solid work on HR1-HR2 structures of CCoV-HuPn-2018 and 2SADS-CoV. The reported results give important structural and atomic detail to the core structure acting as a fusion of the virus and host membranes will trigger future work on elucidating the mode of action after S1 domain attachment to host cells. Experiments have been carried out with care and are adequately described and presented.

Author Response

Thank you so much for your careful check. And  we gratefully appreciate for your valuable comment.

Reviewer 2 Report

Comments and Suggestions for Authors

The manuscript is on crystal structures of fusion cores from CCoV-HuPn-2018 and SADS-CoV.

There are several typos, capital letters, and missing spacing before the references in the text and near tables and figure, that should be corrected. These issues as well as phrases such as “So far, nearly none of very nice therapy can be available to effectively prevent or cure the CCoV-HuPn-2018-related diseases.” (sic) call for a profound proofreading by a native speaker.

The ethics vote is missing which is especially needed when working with data from children. It is not clear where the material and data of the study comes from. This must be outlined in the Methods section.

The title might benefit from an explanation if the abbreviations, also mentioning the country where the study was conducted.

I miss citations for software and ingredients in the Methods section.

Please elaborate and give examples on “jumping into an intermediate species”. It might be good to discuss the interplay with the one health perspective in this respect.

A clear rationale behind the study is missing in the intro section. Also, a limitations section, a conclusion, and practical and theoretical implications of the study finding are missing.

The figure description is incorrect in the text such as Figure B.

The equation below Table 1 should be transferred to the text.

The reference list is incomplete, as only the first authors are mentioned with et al.

In sum, the manuscript needs to be substantially checked for scientific quality and rigor before there is a chance of publishing it in an international journal.

Comments on the Quality of English Language

Extensive editing of English language required

Author Response

Reviewer 2

We thank the reviewer for their thoughtful and helpful comments which have helped improve and strengthen the paper. We have addressed all the raised points and our responses are detailed below.

1 Comment: There are several typos, capital letters, and missing spacing before the references in the text and near tables and figure, that should be corrected. These issues as well as phrases such as “So far, nearly none of very nice therapy can be available to effectively prevent or cure the CCoV-HuPn-2018-related diseases.” (sic) call for a profound proofreading by a native speaker.

1 Reply: We apologize for the mistakes and poor language of our manuscript. We have fixed the problems of typos and capital letters (like in line 97 and 101) and missing spacing before the references in the text and near tables and figure. We have unified the format of unit of volume with mL or μL, substituting ml or μl (like in line 102,124 and 125).

“So far, nearly none of very nice therapy can be available to effectively prevent or cure the CCoV-HuPn-2018-related diseases” has been displaced with “So far, few effective therapies can be applied to prevent or cure the CCoV-HuPn-2018-related diseases”. To further polish our English language, the text underwent further extensive revision in grammar and expression. All the edits are highlighted in yellow.

2 Comment: The ethics vote is missing which is especially needed when working with data from children. It is not clear where the material and data of the study comes fr

Reviewer 2

We thank the reviewer for their thoughtful and helpful comments which have helped improve and strengthen the paper. We have addressed all the raised points and our responses are detailed below.

1 Comment: There are several typos, capital letters, and missing spacing before the references in the text and near tables and figure, that should be corrected. These issues as well as phrases such as “So far, nearly none of very nice therapy can be available to effectively prevent or cure the CCoV-HuPn-2018-related diseases.” (sic) call for a profound proofreading by a native speaker.

1 Reply: We apologize for the mistakes and poor language of our manuscript. We have fixed the problems of typos and capital letters (like in line 97 and 101) and missing spacing before the references in the text and near tables and figure. We have unified the format of unit of volume with mL or μL, substituting ml or μl (like in line 102,124 and 125).

“So far, nearly none of very nice therapy can be available to effectively prevent or cure the CCoV-HuPn-2018-related diseases” has been displaced with “So far, few effective therapies can be applied to prevent or cure the CCoV-HuPn-2018-related diseases”. To further polish our English language, the text underwent further extensive revision in grammar and expression. All the edits are highlighted in yellow.

2 Comment: The ethics vote is missing which is especially needed when working with data from children. It is not clear where the material and data of the study comes from. This must be outlined in the Methods section.

2 Reply: We have substituted the word “children” with “patients” in line 15, 46 and 342. Our paper does not analyze data from children. We just referred to the case reports in literature illustrating identification of CCoV-HuPn-2018 from hospitalized patients, most of whom were children, accordingly. The material and data we used just included the sequences deposited by finders in open database, for example GISAID, which has been outlined in the Methods section 2.1.

3 Comment: The title might benefit from an explanation if the abbreviations, also mentioning the country where the study was conducted. 

3 Reply: Thanks for this valuable comment. We considered to change the title to “Crystal structures of fusion cores from canine corona-virus-human pneumonia-2018 and swine acute diarrhea syndrome coronavirus isolated from Malaysia and China”. But it seemed tedious and too long. So we ultimately decide to keep the original title “Crystal structures of fusion cores from CCoV-HuPn-2018 and SADS-CoV” and the short names of these two viruses get more and more received and conventional. We apologize for politely disagreeing with this comment.

4 Comment: I miss citations for software and ingredients in the Methods section.

4 Reply: Thanks for this good comment. We have added citations associated with beamline and HKL3000 program in line 131 and 132.

5 Comment: Please elaborate and give examples on “jumping into an intermediate species”. It might be good to discuss the interplay with the one health perspective in this respect.

5 Reply: We highly appreciate this point. We have given examples of species jump of SARS-CoV-1 and MERS-CoV in line 34-36, 38-42 and 82-84. Further, we add some new phase describing key factors in species jumping to cross the species barrier in the Discussion section from line 337-341 to connect with evidence found in the receptor of CCoV-HuPn-2018 to obtain adaptation and tropism for new host in the below text.

6 Comment: A clear rationale behind the study is missing in the intro section. Also, a limitations section, a conclusion, and practical and theoretical implications of the study finding are missing.

6 Reply: Thanks for this comment and we highly appreciate your meticulous attitude to science. According to the official writing template and guidance at “https://www.mdpi.com/about/article_types”, this journal needs 4 sections in the text, including Introduction, Materials and Methods, Results and Discussion with optical Conclusion section. But this comment is of high value for improving the quality of our paper. So we add a new section of Conclusion from line 390-398 for completeness of the scientific paper.

7 Comment: The figure description is incorrect in the text such as Figure B.

7 Reply: Thanks for this comment. We have changed the description of “Figure A to D in the Appendix A to D” to “Supplementary Figure A to D” and piled them in the Supplementary Materials section instead of Appendix.

8 Comment: The equation below Table 1 should be transferred to the text.

8 Reply: Thanks for this comment,but we respectably disagree. It is a routine listing the equations associated with statistics of CC1/2 and Rwork or Rfree in Table 1 below Table 1.

9 Comment: The reference list is incomplete, as only the first authors are mentioned with et al.

9 Reply: Thanks, fixed.

10 Comment: In sum, the manuscript needs to be substantially checked for scientific quality and rigor before there is a chance of publishing it in an international journal.

10 Reply: For completeness of scientific writing, we add a new paragraph “2.6 Homology model building” in the Materials and Methods from line 139-149. Besides, we check the whole text and unified formats of terminology, units and so on to pursue scientific quality and rigor to look forward to publishing it in Viruses. All the edits are highlighted in yellow.

om. This must be outlined in the Methods section.

2 Reply: We have substituted the word “children” with “patients” in line 15, 46 and 342. Our paper does not analyze data from children. We just referred to the case reports in literature illustrating identification of CCoV-HuPn-2018 from hospitalized patients, most of whom were children, accordingly. The material and data we used just included the sequences deposited by finders in open database, for example GISAID, which has been outlined in the Methods section 2.1.

3 Comment: The title might benefit from an explanation if the abbreviations, also mentioning the country where the study was conducted. 

3 Reply: Thanks for this valuable comment. We considered to change the title to “Crystal structures of fusion cores from canine corona-virus-human pneumonia-2018 and swine acute diarrhea syndrome coronavirus isolated from Malaysia and China”. But it seemed tedious and too long. So we ultimately decide to keep the original title “Crystal structures of fusion cores from CCoV-HuPn-2018 and SADS-CoV” and the short names of these two viruses get more and more received and conventional. We apologize for politely disagreeing with this comment.

4 Comment: I miss citations for software and ingredients in the Methods section.

4 Reply: Thanks for this good comment. We have added citations associated with beamline and HKL3000 program in line 131 and 132.

5 Comment: Please elaborate and give examples on “jumping into an intermediate species”. It might be good to discuss the interplay with the one health perspective in this respect.

5 Reply: We highly appreciate this point. We have given examples of species jump of SARS-CoV-1 and MERS-CoV in line 34-36, 38-42 and 82-84. Further, we add some new phase describing key factors in species jumping to cross the species barrier in the Discussion section from line 337-341 to connect with evidence found in the receptor of CCoV-HuPn-2018 to obtain adaptation and tropism for new host in the below text.

6 Comment: A clear rationale behind the study is missing in the intro section. Also, a limitations section, a conclusion, and practical and theoretical implications of the study finding are missing.

6 Reply: Thanks for this comment and we highly appreciate your meticulous attitude to science. According to the official writing template and guidance at “https://www.mdpi.com/about/article_types”, this journal needs 4 sections in the text, including Introduction, Materials and Methods, Results and Discussion with optical Conclusion section. But this comment is of high value for improving the quality of our paper. So we add a new section of Conclusion from line 390-398 for completeness of the scientific paper.

7 Comment: The figure description is incorrect in the text such as Figure B.

7 Reply: Thanks for this comment. We have changed the description of “Figure A to D in the Appendix A to D” to “Supplementary Figure A to D” and piled them in the Supplementary Materials section instead of Appendix.

8 Comment: The equation below Table 1 should be transferred to the text.

8 Reply: Thanks for this comment,but we respectably disagree. It is a routine listing the equations associated with statistics of CC1/2 and Rwork or Rfree in Table 1 below Table 1.

9 Comment: The reference list is incomplete, as only the first authors are mentioned with et al.

9 Reply: Thanks, fixed.

10 Comment: In sum, the manuscript needs to be substantially checked for scientific quality and rigor before there is a chance of publishing it in an international journal.

10 Reply: For completeness of scientific writing, we add a new paragraph “2.6 Homology model building” in the Materials and Methods from line 139-149. Besides, we check the whole text and unified formats of terminology, units and so on to pursue scientific quality and rigor to look forward to publishing it in Viruses. All the edits are highlighted in yellow.

Round 2

Reviewer 2 Report

Comments and Suggestions for Authors

I thank the authors for improving their manuscript. Adding a section on the limitations of the study might enhance the balanced discussion on the overall findings. Fixing formating issues would increase the readability of the manscript.

Comments on the Quality of English Language

Minor editing of English language required

Author Response

Thanks for this valuable comment. We have added  this new section named "Limitation of this study" below Conclusion Section from line 399-406 in the revised version.